# Cancer-Selective Treatment of Cancerous and Non-Cancerous Human Cervical Cell Models by a Non-Thermally Operated Electrosurgical Argon Plasma Device

**DOI:** 10.3390/cancers12041037

**Published:** 2020-04-23

**Authors:** Lukas Feil, André Koch, Raphael Utz, Michael Ackermann, Jakob Barz, Matthias Stope, Bernhard Krämer, Diethelm Wallwiener, Sara Y. Brucker, Martin Weiss

**Affiliations:** 1Department of Women’s Health Tübingen, Eberhard-Karls-University Tübingen, 72076 Tübingen, Germany; lukas.feil@student.uni-tuebingen.de (L.F.); Andre.Koch@med.uni-tuebingen.de (A.K.); Bernhard.Kraemer@med.uni-tuebingen.de (B.K.); Diethelm.Wallwiener@med.uni-tuebingen.de (D.W.); sara.brucker@med.uni-tuebingen.de (S.Y.B.); 2Fraunhofer Institute for Interfacial Engineering and Biotechnology, 70569 Stuttgart, Germany; raphael.utz@t-online.de (R.U.); michi-acker@hotmail.de (M.A.); jakob.barz@igb.fraunhofer.de (J.B.); 3Department of Gynecology and Gynecological Oncology Bonn, Friedrich-Wilhelms-University Bonn, 53127 Bonn, Germany; matthias.stope@ukbonn.de; 4Natural and Medical Sciences Institute (NMI), 72770 Reutlingen, Germany

**Keywords:** non-thermal plasma, high frequency electrosurgery, plasma treatment, cold atmospheric plasma (CAP), free radicals, reactive species, cancer selectivity, cervical cancer treatment, cervical intraepithelial neoplasia

## Abstract

Cold atmospheric plasma (CAP) treatment is developing as a promising option for local anti-neoplastic treatment of dysplastic lesions and early intraepithelial cancer. Currently, high-frequency electrosurgical argon plasma sources are available and well established for clinical use. In this study, we investigated the effects of treatment with a non-thermally operated electrosurgical argon plasma source, a Martin Argon Plasma Beamer System (MABS), on cell proliferation and metabolism of a tissue panel of human cervical cancer cell lines as well as on non-cancerous primary cells of the cervix uteri. Similar to conventional CAP sources, we were able to show that MABS was capable of causing antiproliferative and cytotoxic effects on cervical squamous cell and adenocarcinoma as well as on non-neoplastic cervical tissue cells due to the generation of reactive species. Notably, neoplastic cells were more sensitive to the MABS treatment, suggesting a promising new and non-invasive application for in vivo treatment of precancerous and cancerous cervical lesions with non-thermally operated electrosurgical argon plasma sources.

## 1. Introduction

Despite the development of new screening and treatment strategies for early and advanced stages of cervical cancer, patients often suffer for the rest of their lives from radical tumor resections and poorly tolerated systemic therapies. Cervical cancer (CC) and its precursor cervical intraepithelial neoplasia (CIN) are most frequently caused by a persistent infection with human papillomavirus (HPV). Despite the successful introduction of HPV vaccines, cervical cancer is still the fourth most common cancer in women worldwide, with an incidence of 6.6% and a mortality of 7.5% of all the cancer cases combined, according to the latest data from GLOBOCAN in 2018 [1]. There is an urgent need for low-invasive, efficient, and easily applicable treatment options, ideally without the necessity of general anesthesia and in-patient care.

Cold atmospheric plasma (CAP) treatments have offered very promising opportunities for wound healing and antiseptics. Moreover, CAP indicated promising anti-neoplastic effects on several tumor entities, e.g., melanoma, glioma, pancreatic, and several gynecological tumors, particularly breast cancer, ovarian cancer, and cervical cancer [2,3,4,5,6,7,8,9,10,11,12,13]. CAP treatment led to sufficient inhibition of cancer cell growth, interestingly, without tissue swelling, inflammation, or pain. Growing evidence points towards reactive oxygen and nitrogen species (RONS) as being primarily responsible for CAP-triggered cell mechanisms and cell death [13,14,15,16]. Other highly reactive components of CAP include diverse charged particles, free radicals, and ultraviolet and infrared radiation [17]. Today, the most commonly used CAP sources in clinical and research settings are dielectric barrier discharges (DBD) or atmospheric pressure plasma jets (APPJ), which have been mainly developed for wound treatment of the skin [18]. The use of CAP sources in oncologic indications is still limited, and the anatomy of the female genital organs constitutes a problem of accessibility for most of the available, conventional devices. Especially in gynecological oncology, non-thermally operated electrosurgical argon plasma sources could be a suitable alternative for the treatment of precancerous and cancerous lesions due to the small size and high flexibility of the application probes. In this study, we utilized a non-thermally operated Martin Argon Beamer System (MABS) to perform a non-thermal plasma application. The general ability of MABS for the non-thermal treatment of human tissue was shown by dynamic treatment of freshly prepared human preputial tissue samples (Appendix A). Therefore, MABS could be suitable for the cytotoxic in vivo treatment of precancerous and cancerous lesions of the female genital tract.

The purpose of this study was to investigate RONS-driven effects of a non-thermally operated MABS on metabolism and cell survival of different established CC cell lines and a primary non-cancerous cervical tissue (NCCT) cell line established in this study. Therefore, cells were treated with different modes of plasma by varying both the energy and the treatment time. Cellular effects on survival and metabolic function following MABS treatment were identified by growth and cytotoxicity assays, such as MTT and proliferation assays. A specific RONS scavenger was used to investigate the impact of reactive species.

Our data suggest that MABS treatment results in antiproliferative cellular effects on both cervical cancer (CC) cell lines and NCCT cells in a dose-dependent manner. Remarkably, NCCT cells showed significantly less sensitivity to MABS treatment when compared to CC cells.

## 2. Results

In this study, we investigated the effects of a non-thermally operated MABS on CC cell lines as well as on NCCT cells (obtained from a patient undergoing surgery of the cervix uteri at the Department of Women’s Health of the Eberhard-Karls-University Tübingen). Special attention was given to the effects of reactive MABS components on cell proliferation and the metabolic activity of cells.

### 2.1. Assessment of a Non-Thermally Operated MABS by Infrared Thermography and Spatially Resolved Optical Emission Spectroscopy (OES) Measurement by Using an Integrating Sphere

To investigate the cytotoxic impact of MABS during non-thermal plasma treatment, we initially assessed possible thermal effects of MABS treatment within the following experimental setup. First, 100 µL of DMEM was statically MABS treated for 5, 10, and 20 s at 40 W in a 96-well cell culture plate and at a distance of 7 mm. Static treatment of DMEM was immediately followed by infrared thermography and showed no increase of the DMEM temperature after treatment (Figure 1a). Infrared thermography enabled an accurate measurement of the DMEM surface temperature after the MABS discharge and thus expressed the assumed heat transfer into the liquid volume. MABS discharge on cell culture medium, characterized by a filamented discharge of 3–5 mm in diameter, is shown in Figure 1b. To characterize the spontaneous and induced emission spectra of MABS being emitted by excited species, we used conventional OES (Figure 1d). Due to spectral differences depending on the axial position of the OES measurement within the MABS effluent, we additionally performed a spatially-resolved OES by using an integrating sphere [19] (Figure 1c,e). This enabled an absolute and uniform identification of MABS-induced emission spectra. Figure 1d shows an ambient air OES of the MABS effluent recorded at a defined distance of 7 mm from the nozzle. OES peaks in the VIS/NIR region (700–850 nm) were mainly represented by excited argon atoms, whereas no emission was detectable within the VIS spectrum (400–700 nm). Moreover, three lines of nitrogen emission were detected in the ultraviolet (UV)-A region (330–390 nm) and one significant emission of OH at 309 nm in the UV-B range. No emission could be detected in the UV-C range (200–280 nm). Both, the emissions of OH and nitrogen atoms in the UV-range showed relatively low intensities, compared to the argon emission of the visible/near-infrared (VIS/NIR) region. By spatially resolved OES using an integrating sphere, a similar emission and distribution of excited atoms and molecules was observed compared to ambient air OES (Figure 1e). However, the emissions in the VIS/NIR region and especially those of the UV-range could be detected at much higher (up to 15-fold) intensities as well as at better resolutions by using spatially resolved OES. Notably, even with higher sensitivity and resolution of spatially resolved OES, no emission in the UV-C range could be identified.

### 2.2. MABS Treatment of Cervical Cancer Cell Lines Shows Energy and Time-Dependent Reduction of Cell Proliferation

Since MABS treatment might be of high value for the treatment of cervical neoplasia, such as CIN, we characterized its effect on a cellular level. We used a panel of four cervical cancer cell lines as well as healthy primary NCCT cells. To our knowledge, this is the first in-depth characterization of cellular effects of a non-thermally operated MABS in this type of disease. The chosen cell line panel consisted of four cell lines that represented a heterogeneity of cervical cancers, including origin, HPV, and mutational status. The majority of cancerous lesions of the cervix are squamous epithelial carcinomas, which in this study were represented by SiHa (primary tumor) and CaSki (metastatic tumor) cells. These two cell lines are also positive for HPV (type 16 and 18, respectively), whereas DoTc2 4510 and C-33 A cells, representing the adenocarcinomas, are not. An in-house, isolated, and established cell line of non-cancerous cervical primary tissue (NCCT) served as a control. The NCCT cells resemble the phenotype of fibroblasts (Appendix A) and show positive expression of the fibroblast marker fibronectin, whereas they are negative for the expression of the epithelial marker cytokeratin (Appendix A).

To determine the time- and energy-dependent effects of MABS treatment on cell proliferation of CC cells and NCCT, we treated cells under standardized conditions with indicated energies and durations and analyzed their proliferation potential by crystal violet staining six days post treatment (Figure 2). Generally, all CC cell lines and NCCT cells were sensitive to non-thermal MABS treatment. However, the effects observed were clearly dose-dependent, shown by different ED_50_ doses (in Watt) necessary to cause 50% of overall cellular response (Table 1). Throughout CC cell lines, the squamous cancer-derived CaSki cells showed the highest sensitivity for MABS treatment, whereas SiHa, C-33 A, and DoTc2 cells revealed similar antiproliferative growth pattern. Interestingly, these cells were characterized by considerable resistance to MABS treatment up to dosages of 10 s treatment at 10 W, whereas higher energy levels were followed by a significant decrease of CC cell proliferation. Healthy NCCT showed a sensitivity of about 20% at low MABS dosages. Compared to CC cells, NCCT revealed higher resistance to MABS treatment at increasing energy levels. Due to this, the range of 30–60 W and up to 10 s of treatment was found to be a suitable therapeutic window for the treatment of cervical neoplasia, showing strong reductions of CC cell proliferation and only minor effects on the proliferation of NCCT (for the full data set of all cell lines, including all Watt powers and treatment times, please see Appendix A).

### 2.3. Metabolic Activity of CC Cells and Healthy NCCT Cells after Non-Thermal MABS Treatment

After seeing the antiproliferative effect of MABS in our long-term proliferation assay, we aimed at investigating if the antiproliferative MABS effect is associated with an impact on the metabolic activity of cells. As a readout, we used the sensitive and reliable MTT assay to measure the cells’ capability to perform NADH reduction processes. First, SiHa and NCCT cells were treated with different dosages of MABS and were analyzed after 24 h (Figure 3a).

We showed that the metabolic activity of the cells was firmly and dose-dependently decreased in SiHa cells. After 10 s MABS treatment at 10 W, SiHa cells already showed a substantial decrease in metabolic activity compared to healthy NCCT control cells (SiHa: 25 ± 18%; NCCT: 102 ± 41%; Figure 3). The MABS dosage of 25 W for 10 s again showed a significant difference of SiHa and NCCT cells, with NCCT being less sensitive (SiHa: 8 ± 1%; NCCT: 67 ± 3%; Figure 3). However, 20 s of MABS treatment with 60 W significantly attenuated cellular activity in both the cervical cancer cell line SiHa and healthy NCCT with marginal, though significant, differences between the two cell types.

Interestingly, the metabolic activity of SiHa cells was obviously decreased when treated for 10 s of MABS 10 W. However, this was not reflected by a decreased SiHa cell number at the same parameters (Figure 2b). To correlate long-term MABS effects on cell metabolism and proliferation, we performed an MTT assay on SiHa cells over 72 h (Figure 3b). Notably, following a significant decrease after 24 h, we found a complete restoration of metabolic activity, suggesting that the immediate impact on metabolic activity with the respective parameters was not sufficient to significantly decrease cell growth.

Both the proliferation and metabolic activity assays suggest that NCCT cells are, overall, less sensitive to MABS treatment. Moreover, dosage has a major impact on the resulting cell effects.

### 2.4. RONS in MABS-Driven Cell Growth Inhibition

RONS are known to be one of the most critical factors for CAP-based cancer cell growth inhibition and cytotoxicity. Therefore, we supplemented the culture medium with various concentrations of the RONS scavenger N-acetyl-L-cysteine (NAC) [13] and investigated its effects on the proliferation potential of cells in combination with MABS treatment. As treatment parameters, we used 50 W for 10 s, which corresponds to the double ED_50_ dose measured for SiHa cells (see Table 1). The supplementation of NAC into the culture medium prior to MABS treatment caused a significant decrease in the sensitivity of SiHa cells when compared to the MABS negative controls at respective NAC dosages (Figure 4a,f). This disabling effect of NAC in SiHa cells was concentration-dependent and reached its maximum at NAC concentrations of 8 mM (1 mM: 34 ± 7% vs. 8 mM: 83 ± 21%). Increasing the NAC concentration to 20 mM increased cell survival upon MABS treatment but had already slight cytotoxic effects (Figure 4a,f). Similar results of a rescuing effect of NAC after MABS-treatment could be seen for the cancer cell line CaSki (no NAC: 5 ± 2% vs. 8 mM NAC: 76 ± 1%) (Figure 4c,f) and DoTc2 4510 (no NAC: 7 ± 1% vs. 4 mM NAC: 48 ± 6%) (Figure 4d,f). Surprisingly, the C-33 A cell line did not show any rescuing effect after NAC addition (no NAC: 10 ± 2% vs. 4 mM NAC: 14 ± 1%) (Figure 4e,f). NCCT cells were, as seen in our prior assays, less sensitive to the MABS treatment even without NAC supplementation (Figure 4b,f; untreated: 100 ± 2% vs. treated: 57 ± 7%). Supplementation of 1 mM NAC significantly abrogated the detrimental effect of MABS on cell survival, almost rescuing it to control levels (untreated: 101 ± 1% vs. treated: 92 ± 4%). Exceeding NAC concentrations of 2 mM led to the decrease of cell survival regardless of treatment and reached its maximum at 20 mM NAC, showing severe toxicity in NCCT cells (untreated: 2 mM = 101 ± 1%; 8 mM = 79 ± 2%; 20 mM = 26 ± 2%). These data show that RONS play a major role in MABS-mediated toxicity. Scavenging by NAC supplementation abrogates their negative effects.

## 3. Discussion

In this study, we investigated the effects of non-thermally operated MABS, an electrosurgical argon plasma device of the first generation, on cell proliferation and metabolism. MABS is a commonly used electrosurgical plasma source with high availability in clinics worldwide. The aim of our investigation was to prove, that, under non-thermal conditions, MABS has the same impact on cell growth of healthy and cancer tissue compared to conventional CAP sources. Throughout the study we (i) characterized the generation of heat on primary human mucosa during static and dynamic treatment procedures and the spatially resolved optical emission of MABS effluent by OES using an integrating sphere, (ii) investigated the energy- and treatment-time-dependent impact on cell growth of a CC cell line panel and NCCT, and (iii) correlated the observations with RONS-dependent effects on cellular metabolic activity.

The generation of radicals within the gas, liquid, and solid interfaces are known to be the main triggers of CAP effects primarily linked to inhibition of cell proliferation and cell death [14,15,16,19]. Interestingly, by using electron spin resonance (ESR) spectroscopy, our research group recently showed that MABS was characterized by an 18-fold higher increase of total spin density generated within 10 s of treatment when compared to that of the CAP device kINPen med [19,20,21]. OH and H radicals significantly dominated the signals of other radicals in MABS-treated solutions, whereas superoxide anion radicals and hydroxyl radicals were the abundantly found reactive species in kINPen. However, kINPen med and MABS feature completely different principles of plasma generation, plasma tissue conduction, and operating parameters. Therefore, drawing conclusions about the biological impact of MABS on cancerous and healthy cells is hardly possible.

Here, we evaluated for the first time the impact of non-thermal MABS treatment in four different CC cell lines, SiHa, C-33 A, DoTc2 4510, and CaSki, as well as healthy primary cells from cervix uteri (NCCT). We found a significant inhibition of cell proliferation as well as reduced metabolic activity, most likely by MABS-generated RONS. This was next indirectly evaluated by the addition of NAC, which is a synthetic precursor of intracellular cysteine and glutathione (GSH) [13]. NAC addition prior to MABS treatment with increasing concentrations significantly prevented CC cells and NCCT cells from MABS-dependent cytotoxicity (Figure 4). However, NAC concentrations exceeding 8 mM had a cytotoxic impact on the cells, shown by decreased cell growth in both MABS treated cells and controls. As a potential target in anti-cancer therapy, intracellular GSH levels have been investigated intensively in different fields of oncology and were shown to allow cancer cells to cope with the oxidative stress caused by their increased metabolism and proliferation rate. In many cancer cells, strongly increased GSH levels are observed compared to non-cancerous cells [22]. The capability of NAC to counteract CAP-mediated apoptosis has been demonstrated on prostate cancer cells and other tumor entities [2,11,13]. Similar to the present study, the incubation with 5 mM NAC sufficed to increase cell growth after MABS treatment, and according to previous work, was likely via intracellular conversion of cysteine into glutathione. According to Yan et al., transmembrane carrier proteins (aquaporins) may play a major role in CAPs’ mechanism of action [16]. The aquaporin subtypes AQP 1 and especially 3 and 8 were suggested to enable the transmembrane transport of reactive species, mainly H_2_O_2_ [23,24,25]. Notably, the CC cell line SiHa also was shown to express high levels of AQP 1, 3, and 8, whereas human fibroblasts were mainly characterized by only AQP 1 expression, due to a relatively small contribution to skin water homeostasis [26]. It could be hypothesized that the selective MABS efficacy between CC cells and NCCT observed in this study, at least partly, is due to the different expression of specific transmembrane carriers and the resulting differences in the amount of transferred RONS. There have been enormous efforts to investigate the selective effect of CAP on benign and malignant cells. Yan et al. reported of 31 investigated cell lines in several studies that showed a remarkable selectivity [27]. However, the number of studies comparing benign and malignant cells of identical histological origin is low. Often, the studies even lack comparable experimental conditions such as differing treatment parameters and different cell culture media, which was avoided in the present study. Indeed, the malignant cell lines and benign NCCT cells we used anatomically originate from the cervix uteri, thus, the different cells are not characterized by complete histological comparability. Cervical cancer is a highly invasive and often low differentiated tumor, strongly involving the benign peritumoral milieu. Therefore, the comparison of epithelial tumor cells and stromal cells such as primary cervical fibroblasts, nevertheless, reflects the aspects of a future in vivo treatment.

Although all four CC cell lines, as well as NCCT cells, were sensitive to MABS treatment, we found energy- and treatment-time-dependent differences between both the different CC cells and especially when comparing CC cells to NCCT (Figure 2; Appendix A). Generally, the CC panel used in this study combines characteristics of squamous epithelial tumors and adenocarcinomas. Two cell lines (SiHa and CaSki) harbor HPV infections, and one cell line was obtained from a metastatic CC lesion. Moreover, the cell lines harbor distinct mutational patterns, including well-known mutations in gynecological cancers, such as p53, BRCA2, or PIK3CA. Interestingly, CaSki cells, derived from a metastatic site, were most sensitive compared to the primary tumor-derived SiHa, C-33 A, and DoTc2 4510. Apart from this, we found no evidence for a distinct factor resulting in increased resistance to MABS treatment, pointing to multifactorial intracellular processes induced by MABS treatment.

Particularly for malignant cell entities, apoptotic cell death is by far the best-described cellular CAP mechanism in the literature [16]. An increasing number of studies, however, reveal that healthy tissue is not as affected by apoptosis as critically as are malignant cells. Based on different apoptosis assays, our group recently showed that apoptosis cannot sufficiently explain the CAP-dependent inhibition of cell proliferation in primary human fibroblasts using an APPJ [19]. In line with this, Liedke et al. showed similar apoptosis-independent cell effects on healthy mouse embryonic fibroblasts after treatment with an APPJ [28]. Gümbel et al. used MABS for the treatment of an osteosarcoma cell line in comparison with 3T3 mouse fibroblasts at 25 W and 2.6 L argon gas flow [29]. At these conditions, 10 and 30 s of MABS treatment caused a significant inhibition of cell growth of both cell types. Interestingly, this effect was much more pronounced in 3T3 fibroblasts compared to osteosarcoma cells. Besides that, there may be crucial differences between the proliferation of human osteosarcoma and cervical cancer. However, the xenogenic comparison of human cancer tissue and mouse fibroblasts might be associated with serious errors. Furthermore, 5 s of MABS treatment had only marginal effects on cell growth of osteosarcoma cells, whereas 5 s of treatment with the CAP source kINPen med (Neoplas tools, Germany) was followed by a significant decrease of metabolic activity [29]. It can be assymed that if the level of oxidative stress triggered by MABS exceeds the threshold that can be managed by the cellular antioxidative system, e.g., GSH, apoptosis cascades will be induced [30]. Reactive nitrogen species (RNS), such as nitric oxide (NO), seem to undertake further and highly orchestrated mechanisms that are responsible for several regulative effects, such as post-translational modification, S-nitrosation, and epigenetic DNA modifications [31]. Notably, all these effects can be tumor-promoting or tumor-suppressing in a dose-dependent manner, followed by proliferation, chemoresistance, angiogenesis and metastasis or apoptosis, anti-angiogenesis, and enhanced chemo-sensitivity. Moreover, computer simulations showed that RNS reveal enhanced permeability through phospholipid bilayers compared to that of reactive oxygen species (ROS) [32]. Considering that elevated RONS levels overstrain the antioxidant resistance of cancer cells, MABS promises to be highly effective, as we found the reactivity of MABS significantly higher compared to the kINPen due to greater excitation of atoms measured by spatially resolved OES and higher levels of generated oxygen-centered radicals analyzed by ESR (Figure 1) [19,21].

To the best of our best knowledge, this is the first study characterizing MABS by OES measurement. Moreover, we used an integrating sphere, also known as an Ulbricht sphere, to enable an absolute and uniform identification of MABS-induced optical emissions as a result of diffuse and multiple reflectance on the inner surface. We visualized detailed MABS-emissions in the UV, VIS, and IR-regions. Within the UV-A/B region, usual N2-peaks were detected, and no emissions were found in the UV-C region (Figure 1). Excessive UV (especially UV-C) radiation, however, would represent nearly intolerable risks for the development of cancer.

In summary, we showed that the antiproliferative impact of non-thermally operated electrosurgical MABS sources on biological tissue was comparable to common CAP sources. An abundance of new CAP sources are introduced every year. The purchase and commissioning of these devices, however, is associated with high costs due to their limited range of clinical applications and the lack of clinical and in-human experience. To date, only a few in vivo studies with relatively low numbers of enrolled patients have been performed on tumor tissue. First investigations on the efficacy of CAP in clinical settings have been made by repeated, direct CAP treatment of locally advanced head and neck cancers in six patients, and of 12 patients within a confirmatory clinical pilot study using a kINPen MED plasma jet [33,34]. The treatments resulted in reduced pain and overall improved quality of life. However, only 2 out of six patients showed a partial tumor remission for several months by an elevated apoptosis rate within the tumor tissue.

Electrosurgical argon plasma sources such as MABS were developed for high-frequency (HF)-based surgery and are mostly used for different thermal procedures in human [35]. Common HF electrosurgical argon plasma sources are widely available. Due to the variety of possible clinical applications, the long-term costs of these devices are relatively low. Massive thermal damage and damage through excessive UV radiation appear unlikely based on the thermographic and spatially resolved OES measurements performed in this study. Therefore, electrosurgical plasma sources, such as MABS, could be a safe, suitable, highly effective, and cost-efficient alternative to perform in vivo plasma applications for the treatment of human mucosa. In particular, the low-invasive, efficient, and easy-to-perform treatment of precancerous and cancerous lesions of the cervix uteri without the necessity of general anesthesia could be feasible with MABS and other comparable systems. Ex vivo investigations showed that the technology mediates the tissue effects through the full thickness of human epithelium of the cervix uteri [36]. Dynamic MABS treatment with continuous motion enabled the non-thermal treatment of human tissue, which is a feasible treatment strategy for future in vivo applications (Appendix A). Next generation electrosurgical argon plasma devices are characterized by increased control and harmonization of tissue treatments. The development of new operating modes for plasma generation as well as assistance systems for applications will further lower potential risks for adverse tissue damage and application errors. These insights combined with the results from this study will enable clinical in vivo evaluation of the MABS technology for the treatment of precancerous and cancerous lesions of the cervix uteri as well as other sites of human mucosa.

## 4. Materials and Methods

### 4.1. Electrosurgical Argon Plasma Source (MABS)

MABS was generated with the Martin Argon Beamer System (MABS, KLSmartin, Tuttlingen, Germany) including the KLSmartin maXium^®^ high voltage generator unit, the maXium^®^ Beamer as an argon plasma module, and a plasma application probe using argon as a carrier gas. The default “argon plasma coagulation mode” was selected, and the system was operated with a nominal power between 10 and 60 W at a defined distance from the tissue of 7 mm. We found that 10 W was the minimal power that was needed for reliable ignition of the plasma beam. The gas flow did not influence beam ignition and was set to 3.0 L min^−1^.

### 4.2. MABS Treatment Setup

The experiments were performed under a biosafety cabinet to create identical experimental conditions, and the neutral electrode was placed on the metal surface of the hood. Then, a metal block was put onto the neutral electrode to serve as an experiment table for the 96-well plate that was then placed on the metal block. This setup was designed to ensure that the electric currents could be safely deduced and to allow the proper ignition of the plasma beam. The MABS application probe was then placed at a 7 mm distance from the surface of the 100 µL cell culture media. In vitro MABS treatment was performed statically with 3 L min^−1^ gas flow. Depending on the experimental setup, the treatment power and treatment time varied between 10 W and 60 W and between 5 s and 20 s, respectively.

### 4.3. Infrared Thermography

The temperature during MABS treatment was measured by infrared thermography using a FLIR P620 infrared camera (FLIR Systems, Wilsonville, OR, USA) with a set emissivity of 0.98 at a distance of 50–150 cm. Measurements were performed at standardized conditions of 24 °C temperature and 44% humidity during the treatment of (i) 100 µL of DMEM in a 96-well cell culture and (ii) the inner layer of n = 6 primary human preputial tissues from male donors, 1–6 years of age of Caucasian and African origin (non-keratinizing stratified squamous epithelium). The sequences were obtained with one frame per second and were analyzed with the FLIR TOOLS software (FLIR Systems). For static MABS treatment, the human preputial tissues were placed on an aluminum plate connected to the neutral electrode and were MABS treated for indicated time points. For MABS treatment at different movement velocities, the tissue samples were placed on an aluminum plate coupled with an actuated linear rail. Dynamic plasma treatment was mimicked by moving the tissue sample below the fixed plasma probe at different defined velocities of the linear rail.

### 4.4. Optical Emission Spectroscopy (OES)

For OES of the MABS effluent in the region between 200 and 850 nm corresponding to ultraviolet (UV), visible (VIS), and near-infrared (NIR) regions, a DongWoo 700 spectrometer (Dongwoo Optron, Gwangju-Si, Korea) and the Andor iStar ICCD (Andor Technology, Belfast, UK) with a grating of 1200 nm and a blaze of 300 nm were used. The spectrum was generated by the accumulation of 20 measurements at the same gain (gain setting: 180; exposure time: 0.05 s) over a wavelength range of 200 to 850 nm using “step and glue” mode. Calibration of the detector was performed by using a xenon calibration lamp (L.O.T.-Oriel, Darmstadt, Germany). During OES measurement, the MABS source was operated at 12 W and an argon gas flow rate of 3 slm in a continuous working mode. OES was either performed in ambient air or within an integrating sphere at ambient air conditions. For the frontal OES at ambient air, the MABS effluent was focused via quartz glass and a convergent lens at a distance of 7 mm. For the spatially resolved OES, a hollow sphere (Ulbricht sphere, Figure 1c was used as previously described [19].

### 4.5. Cell Lines

CaSki (ATCC CRL-1550), DoTc2-4510 (ATCC CRL-7920), SiHa (ATCC HTB-35), and C-33-A (ATCC HTB-31) were purchased from ATCC (ATCC^®^ TCP-1022™, American Type Culture Collection, Manassas, VA, USA). CaSki and SiHa cells are positive for human papillomavirus (HPV) and are derived from squamous cell carcinomas of the cervix uteri, whereas DoTc2 4510 and C-33 A are derived from adenocarcinomas. NCCT cells were isolated from a primary cervical tissue sample after surgical removal at the Department of Women’s Health, University Hospital Tübingen, Germany. Human preputial tissue from donors after circumcision was used for measurements. The cell division time of NCCTs and each cell line used in this study is about 20–30 h. The scientific use of human tissue samples was approved by the institutional review board of the Ärztekammer Baden-Württemberg (ethical vote: F-2012-078) and the medical faculty of the University Hospital Tübingen (ethical vote: 649/2017BO2). Written informed consent was obtained from all patients. Tissue samples were transported in DMEM supplemented with 1% penicillin/streptomycin. To confirm the benign nature and the lack of histological features of an HPV infection of the primary tissue, pathological review of the specimen was performed by a gynecological pathologist at the pathology department of the university hospital in Tübingen.

### 4.6. Cell Culture

All cell lines were cultured with Dulbecco’s Modified Eagle Medium (DMEM, CAT no. 11965092, Thermofisher Scientific, Waltham, MA, USA) media + 1% GlutaMax (CAT No. 35050061, Thermofisher Scientific) + 1% Penicillin/Streptomycin in standard 100 mm TC-treated polystyrene cell culture dishes and were incubated at 37 °C, 5% CO_2_. The MABS experiments were performed on standard TC-treated polystyrene 96-well plates with a volume of 100 µL cell culture medium. Cells were detached using Trypsin-EDTA 0.05% (Thermofisher Scientific, CAT No. 25300054), transferred in solution, and counted manually using a Neubauer’s counting chamber.

### 4.7. Crystal Violet Proliferation Assay

Cells were plated on 96-well plates (2000 per well) with a volume of 100 µL cell culture medium on day 0. MABS treatment was performed on day 1. On day 7, plates were fixed for 10 min with 96% methanol, stained with 0.1% crystal violet, and washed with dH_2_O. Dried plates were scanned (Epson Perfection V800, Epson, Suwa, Japan; Settings: Positive film mode, 600 dpi, saved as TIF-format) and analyzed with ImageJ software (NIH, Bethesda, MD, USA). Cell survival graph preparation and EC_50_ calculations were achieved with GraphPad Prism software (La Jolla, CA, USA). For RONS scavenger experiments, N-acetyl-L-cysteine (NAC) (Sigma Aldrich, St. Louis, MO, USA, CAS no. 616-91-1) diluted in dH_2_O was added to the media prior to MABS treatment.

### 4.8. MTT (3-(4,5-Dimethylthiazol-2-yl)-2,5-Diphenyltetrazolium Bromide) Assay

Cells were plated on 96-well plates (8000 per well) with a volume of 100 µL cell culture medium on day 0. MABS treatment was performed on day 1. Then, 3 h before the indicated time points (4–72 h) the MTT-reagent was applied and incubated. Readout was performed on a Tecan Sunrise Absorbance microplate reader. Data analysis was performed with GraphPad Prism software (La Jolla, CA, USA).

### 4.9. Immunofluorescence

For immunofluorescent staining, cells were fixed with 4% formaldehyde for 5 min at room temperature (RT) and permeabilized with 0.2% Triton-X for 5 min before blocking in 4% fetal bovine serum albumin (BSA) in 1× PBS supplemented with 0.1% Tween (PBS-T) for 1 h. Cells were incubated overnight at 4 °C with primary antibody in PBS-T with 4% BSA, washed three times with PBS-T, and incubated with secondary antibody and DAPI in PBS-T with 4% BSA for 2 h at room termperature. The following antibodies were used: rabbit-monoclonal anti-Fibronectin (F1) (ab32419, Abcam, Cambridge, UK; 1:300), mouse-anti Cytokeratin (CK3-6H5)-FITC (130-080-101, Miltenyi, Bergisch Gladbach, Germany, 1:100), and secondary antibody goat anti-rabbit-Alexa594 (A11012, Molecular probes, 1:400).

## 5. Conclusions

MABS treatment led to cytostatic and cytotoxic effects in CC cell lines, as well as in NCCT cells of the cervix uteri. The cellular effects were dose- and time-dependent. NCCT cells were significantly less responsive to MABS treatment, suggesting a sort of neoplastic specificity with regard to potential in vivo applications. We investigated such effects via proliferation assays, which showed a massive decrease in viable cells following MABS treatment, especially in CC cells. Cytostatic effects were observed with MTT assays, revealing a higher decrease of metabolic activity in CC cells compared to NCCT cells. By NAC-triggered counteraction of the MABS effects, we suggest RONS are the main active components of MABS. This was shown by the neutralization of RONS, both intracellularly as well as in the cell culture media, with NAC, a scavenger of reactive oxygen and nitrogen species. By doing so, we were able to diminish the described effects of MABS on CC and NCCT cells. Our data indicate that MABS is a suitable option for in vivo MABS treatment of precancerous lesions of the cervix and of CC.

## Figures and Tables

**Figure 1 cancers-12-01037-f001:**
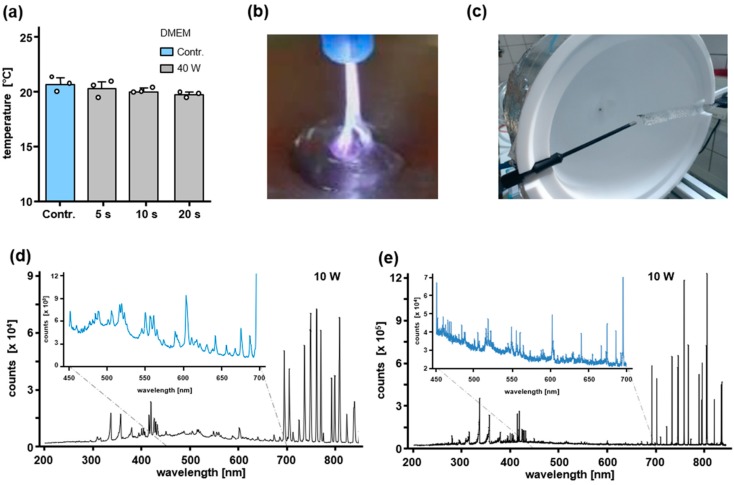
Infrared thermography and OES measurement of the non-thermally operated Martin Argon Plasma Beamer System (MABS). First, 100 µL of DMEM was analyzed during 5, 10, and 20 s of static MABS treatment at 40 W (**a**) in a 96-well cell culture plate. MABS discharge on the DMEM cell culture medium (**b**). For a better illustration, MABS discharge was performed on the surface of a DMEM drop applied to a flat cell culture plastic with the same electric resistance as the multi-well cell culture plate. Results are expressed as the mean ± SD. Setup for spatially resolved OES in a fabricated 100% polytetrafluoroethylene (PTFE) hollow sphere (Ulbricht sphere) (**c**). For conventional (**d**) and spatially resolved (**e**) optical OES of the MABS effluent, the ultraviolet (UV), visible (VIS), and near-infrared (NIR) region were analyzed by accumulation of 20 single OES measurements.

**Figure 2 cancers-12-01037-f002:**
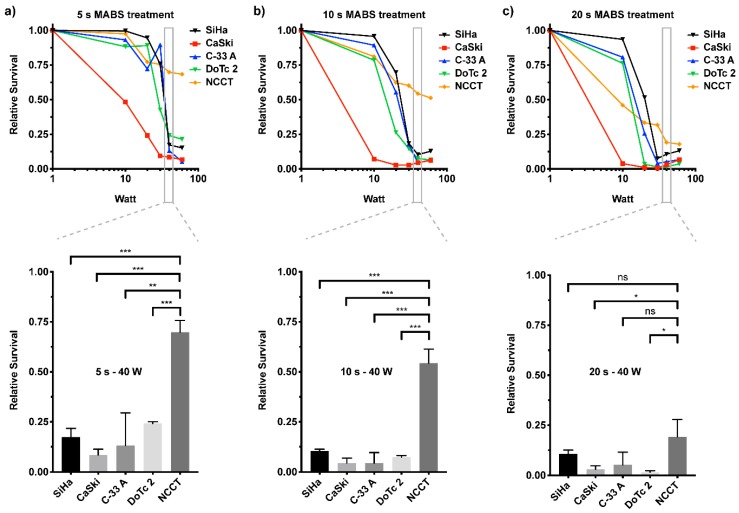
Non-cancerous cervical tissue (NCCT) cells are less sensitive to MABS compared to cervical cancer (CC) cells. Relative survival plots of CC cells (SiHa, Ca Ski, C-33 A, DoTc 2) or NCCT cells fixed six days after MABS treatment with increasing watt power for (**a**) 5 s, (**b**) 10 s, and (**c**) 20 s. Shown is the average of three independent experiments. For better visibility, the standard deviations are excluded from this graph (see Appendix A for full data set). In the bar diagrams (lower part), the values for 40 W and 5 s (**a**), 10 s (**b**), and 20 s (**c**) are plotted. Results are expressed as the mean ± SD of relative survival. * *p* < 0.05, ** *p* < 0.01, *** *p* < 0.001 as determined by Student’s *t*-test.

**Figure 3 cancers-12-01037-f003:**
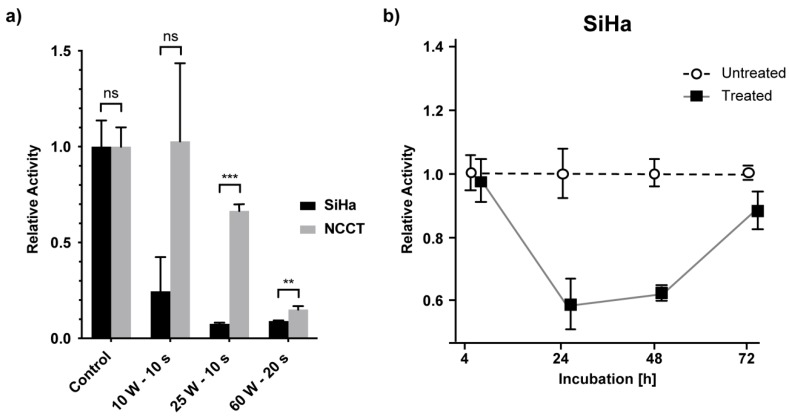
Metabolic activity was decreased in SiHa cells compared to NCCT cells. (**a**) Relative metabolic activity measured via MTT assay in SiHa and NCCT cells 24 h after MABS treatment with given parameters. MABS treatment of SiHa and NCCT cells showed a different impact on metabolic activity. (**b**) Relative metabolic activity normalized to cell numbers of SiHa cells within 72 h after 10 W MABS treatment for 10 s. Metabolic activity was restored after an initial decrease. Results are expressed as the mean ± SD of relative activity. ** *p* < 0.01, *** *p* < 0.001 as determined by Student’s *t*-test.

**Figure 4 cancers-12-01037-f004:**
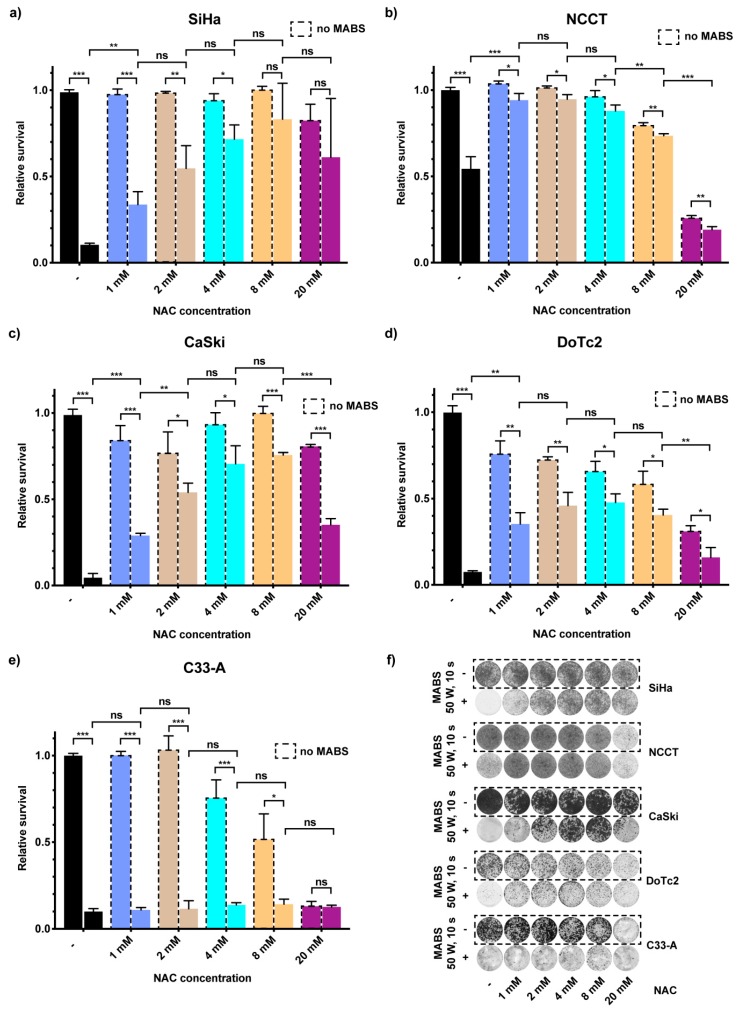
MABS effects are mainly mediated by reactive oxygen and nitrogen species (RONS). Influence of the reactive oxygen species (ROS) scavenger N-acetyl-L-cysteine (NAC) on cell survival of SiHa (**a**), NCCT (**b**), CaSki (**c**), DoTc2 (**d**), and C33-A (**e**) cells incubated in media supplemented with various NAC concentrations prior to MABS treatment for 10 s with 50 W and fixed and analyzed six days after treatment. The bar diagrams show the average result of the proliferation assay of three independent experiments. (**f**) Crystal violet staining of one representative proliferation assay in cell lines analyzed in (**a**–**e**). Results are expressed as the mean ± SD of relative survival. * *p* < 0.05, ** *p* < 0.01, *** *p* < 0.001 as determined by Student’s *t*-test.

**Table 1 cancers-12-01037-t001:** ED_50_ doses (in Watt) necessary to cause 50% of overall cellular response of non-thermal MABS treatment for 10 s.

Cell Line	EC_50_ [W]
SiHa	25.4
CaSki	7.5
C-33-A	21.2
DoTc2 4510	14.1
NCCT	- ^1^

^1^ ED_50_ calculation of NCCT was not feasible.

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
