# Peer review of "Cancer-Selective Treatment of Cancerous and Non-Cancerous Human Cervical Cell Models by a Non-Thermally Operated Electrosurgical Argon Plasma Device"

_cancers, 2020, doi:10.3390/cancers12041037_

Round 1

Reviewer 1 Report

The reviewer thank the authors for taking into account my comments and for improving the quality of their manuscript

Author Response

Thank you for accepting our manuscript for publication in Cancers as well as for your support to improve our work by your primary review.

Reviewer 2 Report

The authors investigated the effect of a non-thermally operated electrosurgical plasma device (MABs) on cancerous and human cells. They performed experiments for physical and chemical characterization of their device, as well as biological experiments to investigate its effect. There are several major items that must be addressed before publication:

  1. The title should be revised to better reflect their study in terms of their work with non-cancerous human cervical tissue. The NCCT cells were isolated from primary cervical tissue, but that does not constitute performing experiments on tissue. Currently, their only work on tissue is included in the introduction and as a supplementary figure without much detail. Furthermore, the tissue they used in the study came from male donors and cannot be described as human cervical tissue. I believe it is more appropriate to remove that statement (“non-cancerous human cervical tissue”) from the title.
  2. Under section 2.1, could the authors explain how they measured temperature and what they mean by average temperature? How does this compare to the local temperature at the point of contact between plasma and the sample? This is necessary for non-specialists to understand the thermal safety of the treatment and central to the theme of their paper.
  3. The legend for Figure 1b states that the discharge was performed on a flat cell culture plastic, while in the picture, it looks like it is performed on water. Could the authors clarify? Also, it is unclear what Figure 1c is and there is no legend to explain this.
  4. There are already reports stating that fibroblasts (their non-cancerous NCCT cells) can tolerate plasma therapy more than epithelial cells (the cancerous cell lines used in their study). This should be included in their results and discussion to avoid false claims of selectivity when talking about plasma therapy, which has been erroneously propagated in this field. The more accurate way of testing selectivity and sensitivity to NTP would be to compare malignant and non-malignant cells of the same cell type.
  5. Under section 2.2 and in Figure 2, it would be helpful if the authors were clear what day the analysis was done, in addition to the statement in the materials and methods.
  6. Under section 2.3 and in Figure 3, how do the authors delineate real decreased metabolic activity compared to just a decrease in viable cells? It could be that metabolic activity is unchanged in live cells, but that their treatment reduced the number of remaining/living cells. Furthermore, it is unclear in Figure 3b what is calculated and graphed as the y-axis ratio is confusing.
  7. Under section 2.4, could the authors describe why 50W for 10s (double ED50) used? As 25W for 10s was the EC50 value, would not adding NAC also benefit overall survival? The statistics of the graph should also be performed to reflect the important comparisons that are relevant to their story. For example, this includes comparing: 1) MABS treatment without NAC vs MABS treatments + NAC and 2) no MABS treatment + NAC vs MABS treatment + corresponding NAC. Figure S4 should also be added into this figure 4, since it includes crucial information and makes their story complete for their cell lines.
  8. Overall, I do not believe their references are adequate. Many primary works and reviews they refer to were from 7 years ago, and there has been significant progress in the field since then. More recent experimental work should be included. Furthermore, there have been several clinical studies with the use of non-thermal plasma for cancers which should be discussed, as this is relevant to their overall goal.

Author Response

Please see attached responses.

Reviewer 3 Report

Accept in present form

Author Response

(The authors gave the same response as above.)

Round 2

Reviewer 2 Report

Authors have addressed the comments and in particular, comment 4 has been very well discussed. No further comments.

Author Response

Reviewer 2, comment 1:

The title should be revised to better reflect their study in terms of their work with non-cancerous human cervical tissue. The NCCT cells were isolated from primary cervical tissue, but that does not constitute performing experiments on tissue. Currently, their only work on tissue is included in the introduction and as a supplementary figure without much detail. Furthermore, the tissue they used in the study came from male donors and cannot be described as human cervical tissue. I believe it is more appropriate to remove that statement (“non-cancerous human cervical tissue”) from the title.

  • We thank Reviewer #2 for his comment. We felt that the expression “Cancer-selective Treatment of Cancerous and Non-cancerous Human Cervical Tissue” would sufficiently reflect our investigation of different confluent cervical cell types as in biology tissue is often defined as ensemble of similar cells and their extracellular matrix. However, to consider the advice of Reviewer #2 we changed the title of the manuscript to “Cancer-selective Treatment of Cancerous and Non-cancerous Human Cervical Cell Models by a Non-Thermally Operated Electrosurgical Argon Plasma Device”

Reviewer 2, comment 2:

Under section 2.1, could the authors explain how they measured temperature and what they mean by average temperature? How does this compare to the local temperature at the point of contact between plasma and the sample? This is necessary for non-specialists to understand the thermal safety of the treatment and central to the theme of their paper.

  • We addressed the points of reviewer #2 with the following rewritten passage under section 2.1 in our manuscript:

“Therefore, 100 µl of DMEM was statically MABS treated for 5, 10 and 20 s at 40 W in a 96-well cell culture plate and at a distance of 7 mm. Static treatment of DMEM was immediately followed by infrared thermography and showed no increase of the DMEM temperature after treatment (Figure 1a). Infrared thermography enabled an accurate measurement of the DMEM surface temperature after the MABS discharge and thus expresses the assumed heat transfer into the liquid volume.”

Reviewer 2, comment 3:

    The legend for Figure 1b states that the discharge was performed on a flat cell culture plastic, while in the picture, it looks like it is performed on water. Could the authors clarify? Also, it is unclear what Figure 1c is and there is no legend to explain this.

  • Indeed, the discharge was performed on DMEM medium without phenol red, which was applied onto a flat cell culture plastic. To clarify this, we rewrote the following passage in the caption of figure 1:

“MABS discharge on DMEM cell culture medium (b). For better illustration, MABS discharge was performed on the surface of a DMEM drop, applied onto a flat cell culture plastic with the same electric resistance of a multi-well cell culture plate.”

  • Figure 1c was explained by the passage “Setup for spatially resolved OES in a fabricated 100% polytetrafluoroethylene (PTFE) hollow sphere (Ulbricht sphere) (c)” at the end of the caption of figure 1. Following Reviewer #2 we put the concerning passage between section b and d of the caption. The caption now reads as follows:

“Figure 1. Infrared thermography and OES measurement of the non-thermally operated MABS. 100 µl of DMEM was analyzed during 5, 10 and 20 s of static MABS treatment at 40 W (a) in a 96-well cell culture plate. MABS discharge on DMEM cell culture medium (b). For better illustration, MABS discharge was performed on the surface of a DMEM drop, applied onto a flat cell culture plastic with the same electric resistance of a multi-well cell culture plate. Results are expressed as the mean ± SD. Setup for spatially resolved OES in a fabricated 100% polytetrafluoroethylene (PTFE) hollow sphere (Ulbricht sphere) (c). For conventional (d) and spatially resolved (e) optical OES of the MABS effluent, the ultraviolet (UV), visible (VIS) and near-infrared (NIR) region were analyzed by accumulation of 20 single OES measurements.”

Reviewer 2, comment 4:

There are already reports stating that fibroblasts (their non-cancerous NCCT cells) can tolerate plasma therapy more than epithelial cells (the cancerous cell lines used in their study). This should be included in their results and discussion to avoid false claims of selectivity when talking about plasma therapy, which has been erroneously propagated in this field. The more accurate way of testing selectivity and sensitivity to NTP would be to compare malignant and non-malignant cells of the same cell type.

  • To cope with the comment of Reviewer #2 we included the following paragraph, discussing previous studies comparing benign and malignant cells. We take note of Reviewer #2’s argumentation that the used cervical cancer cell lines and NCCT cells are not directly comparable. Nevertheless, we see advantages in our study design to simulate a future treatment of neoplastic tissue of the cervix uteri, which we discussed as well.

“There have been enormous efforts to investigate the selective effect of CAP on benign and malignant cells. Yan et al. reported of 31 investigated cell lines in several studies which showed a remarkable selectivity.  However, the number of studies comparing benign and malignant cells of the identical histological origin is low. Often, the studies even lack comparable experimental conditions like differing treatment parameters and different cell culture media, which was avoided in the present study. Indeed, the used malignant cell lines and benign NCCT cells anatomically originate from the cervix uteri, thus the different cells are not characterized by complete histological comparability. Cervical cancer is a highly invasive and often low differentiated tumor, strongly involving the benign peritumoral milieu. Therefore, the comparison of epithelial tumor cells and stromal cells such as primary cervical fibroblasts, nevertheless, reflects the aspects of a future in-vivo treatment.”

Reviewer 2, comment 5:

    Under section 2.2 and in Figure 2, it would be helpful if the authors were clear what day the analysis was done, in addition to the statement in the materials and methods.

  • The analysis was done six days after MABS treatment. As recommended by Reviewer #2 we included this information in section 2.2 also. It is now stated in section 2.2, the caption of figure 2 as well as in the materials and methods section.

Reviewer 2, comment 6:

Under section 2.3 and in Figure 3, how do the authors delineate real decreased metabolic activity compared to just a decrease in viable cells? It could be that metabolic activity is unchanged in live cells, but that their treatment reduced the number of remaining/living cells. Furthermore, it is unclear in Figure 3b what is calculated and graphed as the y-axis ratio is confusing.

  • We are surprised that this point is again raised by the reviewer since we have included a new paragraph addressing this issue in our revised manuscript and we feel this is sufficiently covered by this (as do the other two reviewers).

The concerning paragraph is:
“Interestingly, the metabolic activity of SiHa cells was obviously decreased when being 10 s MABS treated at 10 W. However, this was not reflected by a decreased SiHa cell number at the same parameters (Figure 2b). To correlate long-term MABS effects on cell metabolism and proliferation we performed an MTT assay on SiHa cells over 72 h (Figure 3b). Notably, after a significant decrease after 24 h we found a complete restoration of metabolic activity, suggesting that the immediate impact on metabolic activity with the respective parameters was not sufficient to significantly decrease cell growth.”

  • We agree with the reviewer that our y-axis description is confusing. We changed it to the same y-axis description as in a). The explanation that it is normalized to cell numbers is depicted in the figure legend.

Reviewer 2, comment 7:

Under section 2.4, could the authors describe why 50W for 10s (double ED50) used? As 25W for 10s was the EC50 value, would not adding NAC also benefit overall survival? The statistics of the graph should also be performed to reflect the important comparisons that are relevant to their story. For example, this includes comparing: 1) MABS treatment without NAC vs MABS treatments + NAC and 2) no MABS treatment + NAC vs MABS treatment + corresponding NAC. Figure S4 should also be added into this figure 4, since it includes crucial information and makes their story complete for their cell lines.

  • First of all, we would like to state that the experimental setup, the analysis and the conclusion as well as the presentation of the figure was kept the same from the original submission. We would have preferred that the reviewers suggestions were done in the first review.
  • We have chosen 50W for 10s (double ED50) for our experiment since we have seen in preliminary experiments that this dose looked very convincing for showing the beneficial effect of NAC. In SiHa cells for example, the rescuing effect was from 10% (no NAC) to about 80% (8 mM NAC). Choosing 25W for 10 s (ED50) showed recoveries from 50% to about 90%. So yes, we could have chosen a different treatment parameter (such as 25W for 10s) and would have seen a similar beneficial effect of adding NAC. Since we do describe NAC as a general beneficiary agent, we do not want to specifically explain why one dose of treatment was chosen over the other.
  • In our view, the statistics of the graph sufficiently reflect the relevant points of our study – NAC supplementation is beneficial for the survival of cells after MABS treatment. Adding more statistics to the graph would unnecessarily overload the graph with, in our opinion, non-essential comparisons, and would deviate the attention away from the important ones.
  • For the moment, we kept Figure 4 as it was in our original submission to not overload it with similar looking graphs. If the editor feels we should move the data from the supplement to the main figure we are happy to do so.

Reviewer 2, comment 8:

Overall, I do not believe their references are adequate. Many primary works and reviews they refer to were from 7 years ago, and there has been significant progress in the field since then. More recent experimental work should be included. Furthermore, there have been several clinical studies with the use of non-thermal plasma for cancers which should be discussed, as this is relevant to their overall goal.

  • We disagree in this point. Regarding the scope of our manuscript, we carefully selected the cited investigations of other working groups. In our opinion, the novelty of works and reviews is not the only criteria regarding their significance to the subject of this manuscript.
  • For us, the term clinical study means controlled human in-vivo studies that require both, a positive vote of a legal ethics committee as well as registration in a recognized study register. When searching for the items “Cold atmospheric plasma” or “Cold plasma” or “Non-thermal plasma” or “physical plasma” and “oncology” or “cancer” or “neoplasia” on gov (which is the most important international registration for clinical trials) the only found controlled clinical study on humans is an ongoing study of our own department called “Physical Cold Atmospheric Plasma for the Treatment of Cervical Intraepithelial Neoplasia”. Form our best knowledge this is the first and only controlled and registered clinical IIb pilot study performed in this field. The present study shows several major steps on the way to perform this study and we will be able to present the first in-vivo data soon.

This manuscript is a resubmission of an earlier submission. The following is a list of the peer review reports and author responses from that submission.

Round 1

Reviewer 1 Report

This manuscript by Feil and co-workers presents the results of an in vitro study concerning the ability of non-thermally operated electrosurgical argon plasma, the Martin-Argon-Beamer-System (MABS), to be tuned like“cold plasma” to inhibit the proliferation of cancer cells although preventing toxicity to non-cancerous cells. For that purpose, the authors used 4 different well-established tumor cell lines derived from cervical cancer (CaSki, DoTc2-4510, SiHa and C-33-A) and non-tumor cells isolated from a primary cervical tissue sample after chirurgical removal. Furthermore, the authors also used human preputial tissues from donors after circumcision. The authors essentially based their analysis on two assays, crystal violet colony forming assay and MTT assay, to demonstrate that indeed this argon plasma can selectively kill tumor cells while preserving to some extent non tumor cells. Although, the experiments were well conducted, the results quite clear and the paper well written and easy to follow, I have some reservations about the methodology used (see below).

Major comment: Translating the use of preputial tissues to in vitro cell culture: The aim of using preputial tissue was to demonstrate that the electrosurgical argon plasma can induce heat at the surface of the treated sample if a static position was used while uniform motion of the plasma above the tissue could prevent the generation of critical heat. I regret that the exact physical parameters (energy and treatment time) used for this study were not given. The only given parameters are “a nominal power between 10 and 60 W, a treatment distance of 7 mm and a gas flow rate of 3L/min”. I think this is a very important experiment that was unfortunately under exploited by the authors. Indeed, if the final aim is to use the electrosurgical argon plasma in vivo on cervix tumor (and possibly on other types of solid tumors) then the uniform motion is going to be used during the plasma treatment unlike the static position. Therefore, more investigation regarding the uniform motion could have been done. Related to this observation, the authors then used cancerous and non-cancerous cells seeded in 96-well plates, and covered by cell culture medium, to investigate the anti-cancer capacity of the argon plasma. But then, what is the link between the use of preputial tissue (treated by the argon plasma without being covered by cell culture medium), and cells treated in 96 well plates in static position covered by cell culture medium? Why not used larger wells to allow the use of uniform motion? Why not minimize the volume of cell culture medium to mimic the conditions of argon plasma treatment applied to “preputial tissues”?

Other comments

            First, the authors assessed argon plasma toxicity 6 days post treatment using a crystal violet colony forming assay. They defined a therapeutic window (at an energy around 40 W) in which tumor cells are much more sensitive to the argon plasma than non-cancerous cells (Figure 2). Interestingly, the shape of the curves is very different between cancerous and non-cancerous cells. Almost no response for tumor cells (excepted one, CaSki) up to an energy at which viability dropped drastically while for non-cancerous cells, the shape is smoother. Then the authors used MTT assay that was performed 24h post treatment. They focused on two cell lines, SiHa (cancerous) and NCCT (non-cancerous) (Figure 3). Surprisingly, at energy of 10 W and for a treatment time of 10 s, the relative metabolic activity of SiHa cells is strongly reduced (about 25% compared to 100% in untreated cells) while the colony forming assay showed no sensitivity of this cell line in these operative conditions. Therefore, what is the result of the MTT assay performed exactly in the same conditions as for crystal violet assay (6 days post treatment)? I suspect that the result should reproduce what has been obtained with the colony forming assay, i.e. no or weak sensitivity to plasma treatment. Otherwise, it means that SiHa cells stop producing energy at 24h post treatment but this has no impact on their ability to form colonies afterwards!!!

            What is the volume of cell culture medium used per well in 96-well plates?

            What is the cell division time of each of the cell lines used in this study?

            Figure 1C is not referred in the text body (section 2.1)

            How does MABS plasma discharge look like when the target is a well of 96-well plate filled with cell culture medium? What is the average diameter of the discharge?

            What is the temperature rise of the cell culture medium after 5, 10, and 20s argon plasma treatment in static position at energy of 40 W?

            In the abstract, the authors claim that anti-proliferative and cytotoxic effects are due to the generation of RONS. This conclusion is based on the use of the anti-oxidant NAC. But what is the concentration of plasma-induced H2O2 and NO2- in cell culture medium in the different experimental conditions used in this study (different energies, different treatment times, see Figure 2)? Is H2O2 playing a central role, if any? If NAC is a precursor of intracellular cysteine and glutathione, what is the relative steady-state level of GSH in cancerous versus non-cancerous cells? Could this difference, if any, explain the difference of sensitivity to plasma treatment and the difference in NAC requirement?

            In Figure 4 and Figure S1, the authors show images of crystal violet staining. Although I believe that these images are representative of the cells response to plasma treatment, I’m surprised that they correspond to wells from 96-well plates? Could the authors clarify this point?

            In M&M, the use of spectrophotometer USB2000 is not described

Reviewer 2 Report

In this manuscript, the authors investigated the effects of an argon beam, electrosurgical tool on cancerous and non-cancerous human cervical cells. They performed experiments on both human tissue and cell lines and investigated the physics of their plasma source as well as the biological effects. While the work is interesting, several items must be addressed before publication:

The authors state in their introduction that they performed in vivo treatments of precancerous and cancerous lesions when in fact all reported experiments were in vitro (Page 2, line 65). In Results 2.1., the authors state the tissue was treated with a station plasma source and a dynamic one. It is never stated how the authors were able to achieve the different continuous motions that they claim (10 mm/s―30 mm/s). This should be detailed either in the results or in the materials and methods. It seems strange in both Figure 1 a & b that the variability in temperature would be so large. In some cases, the maximum temperature is almost double the minimum temperature. Can the authors address this? In Results 2.2. the authors used an “in-house isolated established cell line of non-cancerous cervical primary tissue”. There is no reference here, so it is difficult find more information on the cell line. The authors should either add a reference or include more details about the cell line. What type of cell is it? There are publications now that show different cell types respond differently to plasma, and therefore, must be detailed if selectivity is to be claimed. (https://doi.org/10.3390/cancers11091287) Table 1 is currently in Results 2.2 but is not referred to until Results 2.4. Therefore, it should either be moved or referred to earlier. Results 2.3. says that cell viability was assessed, but in fact they measured metabolic activity. Furthmore, the authors interchangeably use viability and metabolic activity when in fact they are different. This should be corrected so their statements more accurately reflect their work. For all experiments with cell cultures, it is not stated if it was treated statically or dynamically with the plasma source. If it was static, in the materials and methods they state that treatment was between 5-20s in which case the temperature would be around 80C. Do the authors measure the temperature of the cell culture media after treatment? Is thermal stress and damage contributing to their observed cellular effects? The authors should provide more details about their treatment and address these questions. Along the same line, the plasma treatment intensity for all cellular experiments were described in Watt, but Figure 1 a &b describe the plasma treatment intensity in time. At what power was the device operated? This is important to draw parallels between temperature and cellular effects. The materials in methods are sorely inadequate and require much more details. In addition to the issues addressed above, it is unclear how much volume is in the 96 wells during treatment; this could provide insight into the temperature of treatment and the species involved, as the larger the volume, the more short-lived reactive species will be lost. Concerning crystal violet, they said the plates were scanned and analyzed without any detail of their instrument or analysis. Furthermore, from the images of their colony forming assay, the cell density seems too high and potentially overlapping. Were the colonies counted by hand or by a software? The discussion is almost a in many cases is just a summary of their results or only scratch the surface of how their work fits into the greater scientific community. For example, on page 8 line 251-255, they talk about their ESR measurements and how it compares with other plasma devices. It would benefit the reader if they go more into detail on the differences: types of species, concentration, biological consequence. Also, as their main point of interest is the clinical accessibility, it would be helpful if they discussed more on the clinical feasibility, application, and challenges. How do they plan to operate the device in the non-thermal mode in the clinic? As a general comment, the authors interchangeably use ‘cold atmospheric plasma’, ‘non-thermal plasma’, ‘cold plasma’, etc. It would help a broad audience of readers to avoid confusion if the authors chose to use one term. 

Reviewer 3 Report

In this manuscript, authors investigated the RONS dependent effects on cell proliferation and metabolism of a non-thermally operated electrosurgical argon plasma source on a tissue panel of cancer cell lines as well as on non-cancerous primary cells of the cervix uteri. This paper is lack of novelty. The data is not good enough to publish in Cancers. Thus, I recommend this paper to be rejected.